# Exploring the Molecular Mechanism of Skeletal Muscle Development in Ningxiang Pig by Weighted Gene Co-Expression Network Analysis

**DOI:** 10.3390/ijms25169089

**Published:** 2024-08-22

**Authors:** Zonggang Yu, Nini Ai, Xueli Xu, Peiwen Zhang, Zhao Jin, Xintong Li, Haiming Ma

**Affiliations:** College of Animal Science and Technology, Hunan Agricultural University, Changsha 410128, China; net236@126.com (Z.Y.); 15291270712@163.com (N.A.); xxl1632501152@163.com (X.X.); zhang1228065671@163.com (P.Z.); jz2725552543@163.com (Z.J.); p13548614212@163.com (X.L.)

**Keywords:** Ningxiang pig, skeletal muscle, candidate genes, molecular mechanism, WGCNA, pathway

## Abstract

With the continuous improvement in living standards, people’s demand for high-quality meat is increasing. Ningxiang pig has delicious meat of high nutritional value, and is loved by consumers. However, its slow growth and low meat yield seriously restrict its efficient utilization. Gene expression is the internal driving force of life activities, so in order to fundamentally improve its growth rate, it is key to explore the molecular mechanism of skeletal muscle development in Ningxiang pigs. In this paper, Ningxiang boars were selected in four growth stages (30 days: weaning period, 90 days: nursing period, 150 days: early fattening period, and 210 days: late fattening period), and the *longissimus dorsi* (LD) muscle was taken from three boars in each stage. The fatty acid content, amino acid content, muscle fiber diameter density and type of LD were detected by gas chromatography, acidolysis, hematoxylin eosin (HE) staining and immunofluorescence (IF) staining. After transcription sequencing, weighted gene co-expression network analysis (WGCNA) combined with the phenotype of the LD was used to explore the key genes and signaling pathways affecting muscle development. The results showed that 10 modules were identified by WGCNA, including 5 modules related to muscle development stage, module characteristics of muscle fiber density, 5 modules characteristic of muscle fiber diameter, and a module characteristic of palmitoleic acid (C16:1) and linoleic acid (C18:2n6C). Gene ontology (GO) enrichment analysis found that 52 transcripts relating to muscle development were enriched in these modules, including 44 known genes and 8 novel genes. The Kyoto Encyclopedia of Genes and Genomes (KEGG) enrichment analysis showed that these genes were enriched in the auxin, estrogen and cyclic guanosine monophosphate-protein kinase G (cGMP-PKG) pathways. Twelve of these genes were transcription factors, there were interactions among 20 genes, and the interactions among 11 proteins in human, pig and mouse were stable. To sum up, through the integrated analysis of phenotype and transcriptome, this paper analyzed the key genes and possible regulatory networks of skeletal muscle development in Ningxiang pigs at various stages, to provide a reference for the in-depth study of skeletal muscle development.

## 1. Introduction

More and more consumers are paying attention to high-quality meat, making the direction of breeding gradually change from high yield to high yield and high quality. Ningxiang pig is an excellent endogenous breed in China, which has the characteristics of roughage tolerance, strong adaptability, delicious meat and rich nutrition [1,2,3,4]. However, its slow growth greatly limits its application and popularization. Skeletal muscle development is an important embodiment of body growth and a key step in the formation of high-quality meat. The LD is the ideal material for studying muscle development because of its large coverage, easy identification and mixed muscle, which can represent the overall development of the body.

Genes can regulate the process of phenotypic changes through time series and tissue-specific expression. Transcriptome sequencing combined with bioinformatics analysis can obtain a large number of candidate genes related to phenotype and potential molecular mechanisms [5]. Some studies have analyzed the molecular mechanism of the Ningxiang pig phenotype by transcriptome sequencing. Jin et al. identified 285 differentially expressed genes by sequencing the whole genome of the LD of the Ningxiang pig combined with differential expression analysis, and further analysis found that they were significantly related to fat deposition and metabolism [6]. Wang et al. sequenced the transcriptome of the subcutaneous fat and intramuscular fat of the Ningxiang pig, and these differentially expressed genes were related to lipid metabolism [7]. Lan et al. analyzed the subcutaneous fat of Ningxiang pigs at different stages by transcriptome and metabolomics, and found that differentially expressed genes regulated the lipid metabolism through cyclic adenosine monophosphate (cAMP) signaling pathway [8]. Liufu et al. sequenced the transcriptome of the LD of Ningxiang pig at different development stages, and the differentially expressed genes were significantly enriched in the pathways of muscle development and intramuscular fat deposition [9]. This research on the Ningxiang pig mainly focused on the molecular mechanism of fat deposition, but the molecular mechanism of muscle development is not enough, and the analytical mechanism is mainly based on differential expression analysis.

WGCNA is a systematic biology method used to study the correlation between gene expression patterns and samples. It can cluster highly correlated genes into modules, and calculate the correlation between module characteristic genes and internal core genes, between modules and sample traits, and the significance of modules, so it is also used to identify candidate markers of tumor and brain development [10]. It has been widely used in the treatment of ovarian cancer [11] and gastric cancer [12], the identification of key genes of muscle color [13] and pectoral muscle development [14], the mining of key genes of different sexes in pig adipose tissue [15] and the identification of the gene network of the lipid metabolism [16]. In order to further analyze the molecular mechanism of muscle development in Ningxiang pigs, in this study, the key genes and signal pathways affecting the muscle development of the Ningxiang pig were mined by whole transcriptome sequencing of the LD of the Ningxiang pig at four development stages combined with WGCNA analysis, in order to provide a reference for the breeding of the Ningxiang pig.

## 2. Results

### 2.1. Module Identification

After sequencing the transcriptome of the LD of 12 Ningxiang boars at different development stages, a total of 12,474 transcripts were obtained by quality control, mapping and expression screening. These transcripts were analyzed by WGCNA, and the results of the module analysis showed that the tree diagram of sample clustering showed that there were no outliers (Figure 1A), and all samples could be used for subsequent analysis. As can be seen from Figure 1B, the soft threshold was 12 when the assumed fitting degree R^2^ between the adjacency matrix and the scale-free network was 0.85. Ten modules of blue, brown, turquoise, gray, yellow, pink, black, green, magenta and red were identified when the combined truncation distance was 0.25 (Figure 1C), and all the screened 9260 transcripts were divided into these 10 modules according to the expression pattern (Figure 1D, Appendix A). Further analysis found that the number of transcripts of each module, blue, brown, turquoise, gray, yellow, pink, black, green, magenta and red modules, respectively, contained 3016, 1672, 3410, 146, 466, 102, 110, 132, 79 and 127 transcripts (Figure 1E). The correlation of each module was further analyzed, and it was found that the brown module had high correlation with the cyan and yellow modules, with correlation coefficients of 0.87 and 0.86, the red module with the green and magenta modules with correlation coefficients of 0.86 and 0.84, the green module with the magenta module with a correlation coefficient of 0.87, and the magenta module with a correlation coefficient of 0.85 (Figure 1F), which indicated that these modules with high correlation might have similar functions.

### 2.2. Screening Skeletal Muscle Development-Related Co-Expression Gene Modules by WGCNA

#### 2.2.1. Development Stage Feature Module and Hub Gene Identification

After screening, 9260 transcripts were obtained, and further analysis based on the screened transcripts found that the optimal soft threshold β was 12. According to the |Cor| > 0.5 and *p* < 0.05 of the module and the muscle growth and development stage, five modules related to muscle growth and development were identified among the 10 modules, except from the gray module. The characteristic modules of the nursery period were green (r = −0.652, *p* = 0.0216), turquoise (r = 0.717, *p* = 0.00868) and yellow (r = 0.763, *p* = 0.00389), and the characteristic modules of the early fattening period were blue (r = 0.647, *p* = 0.023). The characteristic modules in the late fattening stage were green (r = 0.758, *p* = 0.00428), turquoise (r = −0.684, *p* = 0.0142) and brown (r = −0.764, *p* = 0.00382) (Figure 2A). Among them, the green and turquoise modules were characteristic modules in the nursery period and the late fattening period (Figure 2A). Hub genes were screened according to the feature eigengene-based connectivity (kME) > 0.8 and gene significance (GS) > 0.5. A total of 2632 hub genes were identified in three modules of the nursing period. There were 74 hub genes in the green module, 2257 hub genes in the turquoise module and 301 hub genes in the yellow module (Figure 2B,E). Some 819 hub genes were identified in the blue module of the early fattening period (Figure 2C,E). A total of 3100 hub genes were identified in the three modules of the late finishing period, of which 76, 2200 and 824 were in the green, turquoise and brown modules, respectively (Figure 2D,E).

#### 2.2.2. Key Modules of Muscle Fiber-Related Indexes of LD and Identification of Hub Gene

According to the analysis results in Figure 3A (Appendix A), the characteristic module of muscle fiber density was red (r = −0.671, *p* = 0.0169), and the characteristic module of muscle fiber diameter was magenta (r = −0.732, *p* = 0.00635), green (r = 0.757, *p* = 0.00436), red (r = 0.724, *p* = 0.00776), turquoise (r = −0.732, *p* = 0.0068) and brown (r = −0.622, *p* = 0.0308). No module related to muscle fiber type was identified (Figure 3A). A total of 3032 hub genes were identified from 5 characteristic modules of muscle fiber diameter, of which 627, 2256, 83, 47 and 19 were identified in brown, turquoise, green, magenta and red, respectively (Figure 3B,D). A total of 22 hub genes were identified in the red module related to muscle fiber density (Figure 3C,D).

#### 2.2.3. Key Module of Amino Acids and Fatty Acids in LD and Identification of Hub Gene

No characteristic module related to amino acids was identified (Appendix A).

The identification results of the fatty acid characteristic modules show that the characteristic modules of C16:1 and C18:2n6C were all blue modules (Figure 4A, Appendix A). There were 714 hub genes and 824 hub genes in module of C16:1 and C18:2n6C, respectively (Figure 4B–D).

### 2.3. Functional Enrichment Analysis of Characteristic Genes of Muscle Phenotype Related Modules

#### 2.3.1. Functional Enrichment Analysis of Hub Genes of Related Modules in Muscle Development Stage

The hub genes in the early fattening module by GO enrichment analysis significantly enriched 2297 terms, including 60 terms related to muscle development, such as GO:0048661 (positive regulation of smooth muscle cell proliferation), GO:0048634 (regulation of muscle organ development), GO:1901861 (regulation of muscle tissue development), GO:0035914 (differentiation of skeletal muscle cells), GO:0016202 (regulation of skeletal muscle tissue development), etc. (Figure 5). Sixty terms contained 162 transcripts.

The hub genes in the early fattening period by GO enrichment analysis were significantly enriched in 1573 terms, including GO:0010657 (apoptosis process of muscle cells), GO:0014812 (migration of muscle cells), GO:0031032 (actin structure), GO:0055017 (growth of myocardial tissue), and GO:0014909 (smooth muscle cell migration), etc. (Figure 6). These terms contained 70 muscle-related transcripts.

The hub genes in the late fattening period were significantly enriched in 2674 terms. Among them, there were 69 terms related to muscle development, such as GO:0048634 (regulation of muscle organ development), GO:1901861 (regulation of muscle tissue development), GO:0016202 (regulation of striated muscle tissue development) and GO:004866 (regulation of smooth muscle cell proliferation) and so on (Figure 7). Sixty-nine terms contained 239 transcripts.

#### 2.3.2. Functional Enrichment Analysis of Characteristic Genes of Muscle Fiber-Related Modules

To further explore the function of the hub genes related to muscle fiber diameter and muscle fiber density, the GO enrichment analysis was carried out. It was found that the hub genes related to muscle fiber density were significantly enriched in 277 terms, including GO:0032982 (myosin filament), GO:0030016 (myofibril), GO:0006936 (muscle contraction) and GO:0003012 (muscle system process), GO:0016459 (myosin complex) and another 14 terms (Figure 8). There were 12 transcripts in 14 terms.

The hub genes related to muscle fiber diameter were significantly enriched in 2690 terms, including GO:0048634 (regulation of muscle organ development), GO:1901861 (regulation of muscle tissue development), GO:0016202 (regulation of skeletal muscle tissue development), GO:0048661 (positive regulation of smooth muscle cell proliferation) and GO:0048660 (regulation of smooth muscle cell proliferation) and another 68 terms were related to muscle (Figure 9). Sixty-eight terms contained 248 transcripts.

#### 2.3.3. Enrichment Analysis of Characteristic Genes of Fatty Acid-Related Modules in LD

GO enrichment analysis of the hub genes concerning fatty acid-related modules enriched 1573 terms, including 45 terms related to lipids, including GO:0006650 (glycerophospholipid metabolism process), GO:0006644 (phospholipid metabolism process), GO:0046486 (glyceride metabolism process) and GO:0044255 (cellular lipid metabolism process). These fatty acid-related terms contained 276 transcripts (Figure 10).

#### 2.3.4. Overlapping Analysis of Characteristic Genes and Hub Genes of Muscle Phenotype

To further verify the accuracy of the identification of module characteristic genes, the identified muscle phenotype-related characteristic gene and GO-enriched genes related to related muscles were overlapped and analyzed. In the nursery period, 85 common transcripts were obtained (Figure 11A). In the early fattening period, 19 common transcripts were detected (Figure 11B). In the late fattening period, 132 common transcripts were obtained (Figure 11C).

By overlapping the analysis of characteristic genes related to muscle fiber density and hub genes related to muscle development, three common genes were obtained (Figure 10E). A total of 128 common genes were obtained by overlapping the analysis of characteristic genes related to muscle fiber diameter and hub genes related to muscle development (Figure 11D).

After the analysis of gene overlap between characteristic fatty acid-related genes and hub genes related to muscle development, it was found that there were 23 and 26 common genes enriched in C16:1 and C18:2n6C, respectively (Figure 11F,G).

Upset analysis showed that 52 transcripts were the most overlapped phenotypes, and 3 phenotypes were nursery period, late fattening period and muscle fiber diameter. The 52 transcripts contain 44 known genes and 8 novel genes (Figure 11H,I).

KEGG enrichment analysis of 52 transcripts related to skeletal muscle development showed that these genes were significantly enriched in calcium ion, auxin, estrogen, cGMP-PKG, ErbB and other signal pathways (Appendix A).

Compared with 49 transcripts obtained by differential expression analysis [17], WGCNA analysis also identified 36 non-differentially expressed transcripts, including 30 known genes and 6 new genes, and also identified 16 common differentially expressed transcripts, including 14 known genes and 2 new genes in common genes (Appendix A).

### 2.4. Protein–Protein Interaction (PPI) and Transcription Factor (TF) Analysis of Muscle Phenotype-Related Genes

To further analyze whether there was interaction between these common transcripts, protein–protein interaction analysis and transcription factor prediction analysis were carried out. The reliability of the results was further analyzed in mice and human. There were 29, 25 and 29 interacting proteins found in human, pig and mouse respectively. The interaction of 20 proteins (MEF2A: myocyte enhancer factor 2A, FOS: Fos proto-oncogene, AP-1 transcription factor subunit, EGR1: early growth response 1, RBPJ: recombination signal binding protein for immunoglobulin kappa J region, PRKAA1: protein kinase AMP-activated catalytic subunit alpha 1, KLHL40: kelch like family member 40, HDAC2: histone deacetylase 2, LMOD2: leiomodin 2, DDX5: DEAD-box helicase 5, PPP3CA: protein phosphatase 3 catalytic subunit alpha, EGR2: early growth response 2, CREB1: cAMP responsive element binding protein 1, CDKN1B: cyclin dependent kinase inhibitor 1B) were stable. Furthermore, the core protein of Top15 in the interaction proteins of three species was identified by Cytohubba analysis (Figure 12D–F). There were 20, 18 and 19 transcription factors in human, pig and mouse, respectively. 18 transcription factors (MEF2A, EGR1, HDAC2, DDX5, EGR2, CDKN1B, NFATC3: nuclear factor of activated T cells 3, FOS, RBPJ, USF2: upstream transcription factor 2, c-fos interacting, YY1: YY1 transcription factor, CREB1, KLF5: Kruppel Like Factor transcription factor 5, COPS2: COP9 signalosome subunit 2, etc.) were common in three species. There were 11 common genes in set of muscle-related common hub genes, interacting proteins and transcription factors (Figure 12C).

## 3. Discussion

With the development of sequencing technology, exon, micro RNA (miRNA) and Long non-coding RNA (lncRNA) sequencing technologies have been developed one after another [18], which are widely used to identify markers of cancer occurrence and diagnosis, key regulatory molecules in biological processes such as immune response and development [19,20,21,22,23], and have made many convincing progress. These results shows that transcriptome sequencing is a reliable method to explore the genetic mechanism of traits. Gene is the mainly inherent factor that affects phenotype, so it is very important to mine the key genes that affect phenotype. The hypothesis of WGCNA is that genes with the same expression pattern in different samples have similar functions [10], so it is often used to mine new genes in a certain phenotype or new functions of known genes. In this study, the candidate genes related to muscle development in Ningxiang pig were mined by whole transcriptome sequencing combined with WGCNA. 12,474 transcripts were identified by whole transcriptome sequencing of LD in four development stages of Ningxiang pig, including 11,033 known genes and 1441 novel genes. After WGCNA screening, 9260 transcripts were obtained, including 8305 known genes and 955 novel genes. These genes were clustered into 10 expression patterns, corresponding to 10 modules respectively. Further, by combining the modules with phenotypes, it is identified that different characteristic modules correspond to different development stages, among which the nursery period corresponded to green, turquoise and yellow modules, totaling 2967 hub genes, of which turquoise and yellow were positively correlated modules, and green was negatively correlated modules. There were 1227 hub genes corresponding to positive correlation blue modules in the early fattening period, and there were 3579 hub genes corresponding to positive correlation green, negative correlation turquoise and brown modules in the late fattening period. During the nursery period and the late fattening period, there were abundant regulatory signal pathways and a large number of genes involved in regulation. The same module may show opposite correlation in different development stages, which was consistent with the our results that the diameter of muscle fiber increases rapidly in the nursery period and the density of muscle fiber decreases sharply in the late fattening period.

The characteristic module related to muscle fiber density was red, including 22 hub genes. The characteristic modules involved in the development of muscle fiber diameter were magenta, green, red, turquoise and brown, with a total of 3680 hub genes, which showed that the muscle fiber diameter changes frequently in the four development stages after birth. These results were consistent with the significant increase in muscle fiber diameter with growth. The characteristic module of transformation with fast and slow twitch muscle fibers was not identified.

To further study the key genes involved in muscle development, the function enrichment analysis of the hub genes in the module and the venn analysis of each phenotype were carried out. The results showed that 52 genes played an important role in the nursery period, late fattening period and muscle fiber diameter. The 52 genes include 44 known genes such as *MEF2A*, myostatin (*MSTN)*, *CREB1* and *HDAC2*, and 8 novel genes such as MSTRG.35863, MSTRG.31180 and MSTRG.25310. *MEF2A* and *MSTN* in known genes have been proved to be involved in the process of muscle formation and atrophy [24,25,26]. They may also regulate the muscle development of Ningxiang pigs by promoting myogenesis and atrophy. The deficiency of *KLHL40* can cause the instability of filament protein and lead to linear myopathy, which showed that *KLHL40* plays an important role in the normal development of muscle [27]. This study also found that *KLHL40* is a key gene in different stages and the development of muscle fiber diameter, which shows that it plays a key role in the process of muscle fiber diameter increase in Ningxiang pigs. *KLF5* promotes myogenic differentiation by directly targeting myogenic differentiation 1 (*MYOD1)* in the process of muscle differentiation in mice [28]. This study also identified it as a key gene in the process of muscle development, which also suggested that *KLF5* may also regulate the myogenic differentiation of Ningxiang pigs by combining *MYOD1*. Homer scaffold protein 1 (*HOMER1*) was identified as the key gene in this study, and *HOMER2* and *HOMER3* of homer family were proved to be involved in muscle and neuromuscular adaptation during muscle development [29], but whether *HOMER1* is involved in muscle development has not been reported. Therefore, this study broadens the function of *HOMER1*, but its role in muscle development needs further verification. This study found that mitogen-activated protein kinase kinase 6 (*MAP2K6*) is the core gene in muscle development, and it has been reported that *MAP2K6* is the key molecule in the mitogen-activated protein kinase (MAPK) signaling pathway and participates in muscle development. Some studies have found that the expression of *HDAC4* can inhibit the process of muscle differentiation [30], and it was also the hub gene in the muscle development of Ningxiang pigs, so it can be inferred that *HDAC4* may also inhibit the myogenic differentiation of Ningxiang pigs. 

To further analyze the relationship between these genes related to muscle development, PPI and transcription factors were predicted and analyzed. The results showed that there were 18 transcription factors in them, and there were interactions between proteins encoded by 25 genes. In order to verify the stability of these transcription factors and their interactions, we tested them in mice and humans. The results showed that 20 of them had stable interactions, and 18 transcription factors identified in Ningxiang pigs were highly conserved in the other 2 species. This shows that these interactions may also exist in the process of pig muscle development. Whether the functions are consistent needs further verification. It has been found that *HDAC5* may inhibit the activation of ERK/EGR1 signaling to regulate the expression of *MEF2*, thus participating in ventricular remodeling [31]. This relationship between HDAC–EGR1–MEF2A has been verified in ventricular remodeling, and whether this interaction can be extrapolated to the muscle development of Ningxiang pigs needs further verification. It was reported that *CREB* nuclear transcription factor can stimulate the expression of c-Fos and then affect brain development [32], and the interaction between CREB and FOS was verified here. This interaction may also exist in the muscle development of Ningxiang pigs, and their influence on muscle development needs further verification. Other interactions are involved in the process of muscle development and need to be further explored.

## 4. Materials and Methods

### 4.1. Experimental Animals and Sample Preparation

Twelve healthy male Ningxiang pigs were selected, including three Ningxiang pigs at 30, 90, 150 and 210 days after birth. The pigs at the same stage were full-sibs and half-sibs at different stages. The experimental pigs came from the pig farm of Ningxiang Dalong Animal Husbandry Technology Co., LTD (Changsha, China). All pigs in the study were provided ad libitum access to water and feed. LD muscle of the fourth to last ribs was collected within 30 min after slaughter. The muscle samples were frozen in liquid nitrogen and then transferred to −80 °C for storage until RNA extraction.

### 4.2. Phenotypic Detection

#### 4.2.1. Detection of Type Proportion, Diameter and Density of Muscle Fibers

The LD was fixed with 4% paraformaldehyde and embedded with paraffin within 3 days. The embedded wax block was cut into 4 μm and 10 μm slices for subsequent detection of muscle fiber type, diameter and density. The proportion of muscle fiber types was detected by immunofluorescence staining, and the method was referred to Yu. et al. [17].

The detection process of muscle fiber diameter and density is as follows: 4 μm paraffin slices are baked in water, dried and stained with hematoxylin-eosin. Photos of stained sections were collected by tissue section digital scanner, and the proportion, diameter and density of muscle fiber types were determined by CaseViewer v2.4 and imageJ v4.9 software.

#### 4.2.2. Detection of Fatty Acids and Amino Acids

Determination of hydrolyzed amino acids in LD was achieved by accurately weighing 0.5 g of fresh meat sample, adding 10 mL of HCl (6 mol/L) and putting it in a thermostat at 110 °C for 22 h. After cooling, the hydrolysate was filtered, rinsed with deionized water, and the volume fixed to 50 mL. An extraction of 1 mL of filtrate was made, it was dried to remove the solvent, and finally dissolved with 1 mL of 0.01 N HCl, passed through a 0.22 μm filter, and the filtrate applied to an amino acid automatic analyzer (Hitachi LA8080, Tokyo, Japan). All required chemical reagents were purchased from sigma (St. Louis, MO, USA). The data were collected from OpenLAB CDS 2 software. The formula for calculating the content of hydrolyzed amino acids in the sample is as follows:$$X_{i}=\frac{c_{i}\times 50\times F\times V\times M}{m\times 10^{9}}\times 100$$
where X: the content of amino acids in the sample, in grams per hundred grams (g/100 g); c_i_: amino acid content in the sample determination solution, with the unit of nanogram per 20 microliters (ng/20 μL); 50: amino acid content in the sample determination solution, in ng/mL, *F*: sample dilution multiple; *V*: constant volume after hydrolysis, in mL; *M*: sample mass, in grams (g); 10^9^: to convert the sample content from nanograms (ng) to the coefficient of a gram (g); 100: the coefficient to convert the sample content from grams (g) to hundreds of grams (100 g).

Determination of fatty acids in LD was performed as previously described [33], by weighing 0.5 g of freeze-dried and crushed muscle samples, adding benzene-petroleum ether solution for leaching for 24 h, then adding potassium hydroxide-methanol solution for saponification and methyl esterification to generate fatty acid methyl ester, which was analyzed by capillary column gas chromatography (SRI, Palo Alto, CA, USA). The data were collected from peak simple software. The content of a given component was calculated by measuring the percentage of the corresponding peak area vs. the peak area of all components. The calculation formula is as follows:$$Y_{i}=\frac{A_{Si}\times F_{FAME_{i}-FA_{i}}}{\sum A_{Si}\times F_{FAME_{i}-FA_{i}}}$$
where *Y_i_*: the percentage of a certain fatty acid in the sample to the total fatty acid, %; *A_Si_*: the peak area of each fatty acid methyl ester in the sample determination solution; *F_FAMEi-FAi_*: the coefficient of fatty acid methyl ester *i* converted into fatty acid; ∑*A_Si_*: the sum of peak areas of fatty acid methyl esters in the sample determination solution.

### 4.3. RNA-Seq

RNA-seq data were obtained from our previous research (Access numbers: SRR14211782-SRR14211784 (30 days after birth), SRR14211772-SRR14211774 (90 days after birth), SRR14211767-SRR14211769 (150 days after birth), SRR1421792-SRR1421794 (210 days after birth) [17].

### 4.4. Bioinformatics Analysis

#### 4.4.1. Sequencing Data Sorting and Trimming

The adapters in the reads from the offline raw data were removed, and the reads without inserting fragments due to the self-connection of adapter were removed. The bases with low quality (quality score < 20) at the end of the sequence (3′ end) were trimmed. Then those remaining sequences with the quality score less than 10 were eliminated. The reads with N content exceeding 10% and sequences with length less than 20 bp after adapter removal were discarded. The raw data were filtered by Fastp (v0.17) software [34], and the clean data were statistically analyzed by SeqPrep (V1.2) (https://github.com/jstjohn/SeqPrep, accessed on 13 August 2020) and Sickle (V1.33) (https://github.com/najoshi/sickle, accessed on 13 August 2020).

Clean data was mapped to the reference genome of Ningxiang pig by Hisat2 (v2.2.1) software [35,36]; the compared data were assembled with StringTie (v2.2.0) [37], and a new transcript was obtained with GFFcompare (v0.12.6) [38] software.

Samtools software (v2.6.4) was used to compare, format and sort the results [39]; Htseq software (v0.11.3) was used to count clean reads [40]. RSEM software (v1.3.5) was used to calculate TPM value of mRNA [41].

#### 4.4.2. WGCNA Analysis

Firstly, 12,474 transcripts (Appendix A) of Ningxiang pigs in four developmental stages were screened. The screening criteria were TPM > 1, coefficient of variation was 0.1, scale-free distribution and soft threshold were set to R^2^ > 0.85 to determine the soft threshold (β power). The parameters of module identification were set as follows: the minimum module size was 30, the lower limit of kME of the correlation between the expression pattern of module genes and module characteristic genes was set to 0.3, the module merging distance was set to 0.25, and the network type was directed network. Modules with module correlation coefficient |r| > 0.5 and *p* < 0.05 were considered as stage-specific modules. The criterion for identification of hub genes was gene significance (GS) > 0.5, and the correlation coefficient between modules and characteristic genes was greater than 0.8.

#### 4.4.3. Enrichment Analysis of GO and KEGG

Gene ID was converted by bioDBnet website (https://biodbnet-abcc.ncifcrf.gov/, accessed on 13 August 2020). Firstly, AnnotationHub R package was used to obtain the annotation file of Ningxiang pig’s genes, and then clusterProfiler R package was used to perform GO enrichment analysis for the screened transcripts. Then, KEGG pathway enrichment analysis was carried out for transcripts/genes significantly enriched in GO terms via KEGG Orthology-Based Annotation System (KOBAS) website [42].

#### 4.4.4. Prediction and Analysis of Transcription Factors

According to the domain information contained in the products of gene transcription, transcription factors and transcription factor families can be predicted. Transcription factors were identified by AnimaltfDB 4.0 (https://guolab.wchscu.cn/AnimalTFDB4//#/, accessed on 13 October 2023).

#### 4.4.5. Protein–Protein Interaction Analysis

The protein–protein interaction network of the gene of interest in human, mouse and pig was analyzed by the STRING database (http://string-db.org/, accessed on 13 October 2023). The interaction network is visualized by cytoscape software (v4.9.1), and the key genes are screened according to MCC algorithm in Cytohubba plug-in.

## 5. Conclusions

The characteristic modules and hub genes of the Ningxiang pig’s development stages, muscle fiber diameter, muscle fiber density and muscle fiber type ratio were identified by transcriptome sequencing combined with WGCNA analysis. These hub genes, as transcription factors or regulatory proteins, participated in the regulation of skeletal muscle development by enriching the signaling pathways for the growth hormone, estrogen and cGMP-PKG. These results provide a reference for the further study of skeletal muscle development in pigs.

## Figures and Tables

**Figure 1 ijms-25-09089-f001:**
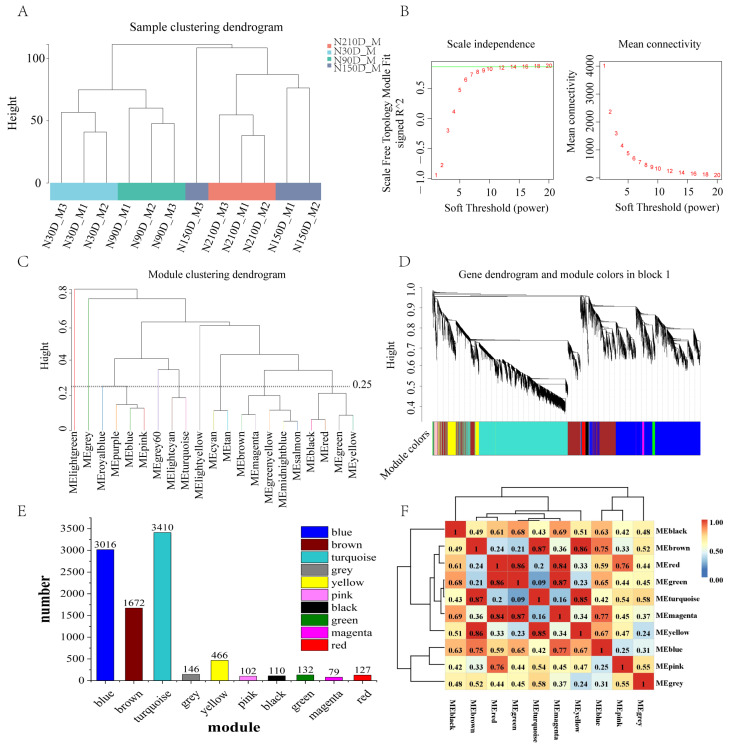
WGCNA analysis of transcript of LD in Ningxiang pig. (**A**) Shows the sample clustering tree, (**B**) the scale-free topology fit curve and mean connectivity curve, (**C**) the combined truncation distance tree with truncation distance 0.25, (**D**) the transcript module classification tree, (**E**) the module member statistical bar chart, (**F**) the module correlation clustering heatmap.

**Figure 2 ijms-25-09089-f002:**
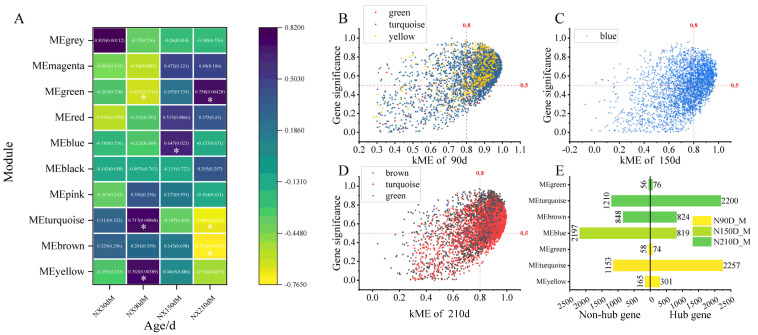
Identification of characteristic modules and hub genes in developmental stages of Ningxiang pigs. (**A**) Shows the heatmap of correlation with modules in different developmental periods, purple indicates positive correlation, yellow indicates negative correlation; (**B**–**D**) show the scatter plots of hub gene screening in nursery period, early fattening period and late fattening period, respectively, in which the X axis was the connectivity of characteristic gene modules and the Y axis was the gene significance. (**E**) The summary histogram of hub gene, the characteristic module of the developmental period. * Indicates that the significance of correlation coefficient is *p* < 0.05.

**Figure 3 ijms-25-09089-f003:**
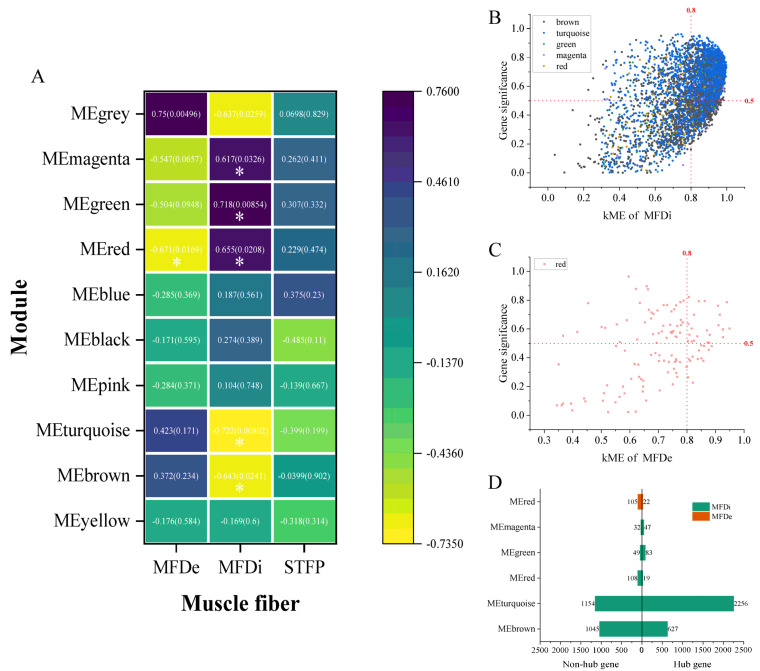
Identification of characteristic modules and hub genes about muscle fibers. (**A**) Correlation heatmap between muscle fiber diameter, muscle fiber density and slow-twitch fiber ratio and modules, purple indicates positive correlation, yellow indicates negative correlation. (**B**,**C**) Hub gene screening scatter plot of muscle fiber diameter and muscle fiber density, in which the X axis was the connectivity of characteristic gene modules and the Y axis was the gene significance. (**D**) Summary histogram of genes in the characteristic module of muscle fiber. * Indicates that the significance of correlation coefficient is *p* < 0.05.

**Figure 4 ijms-25-09089-f004:**
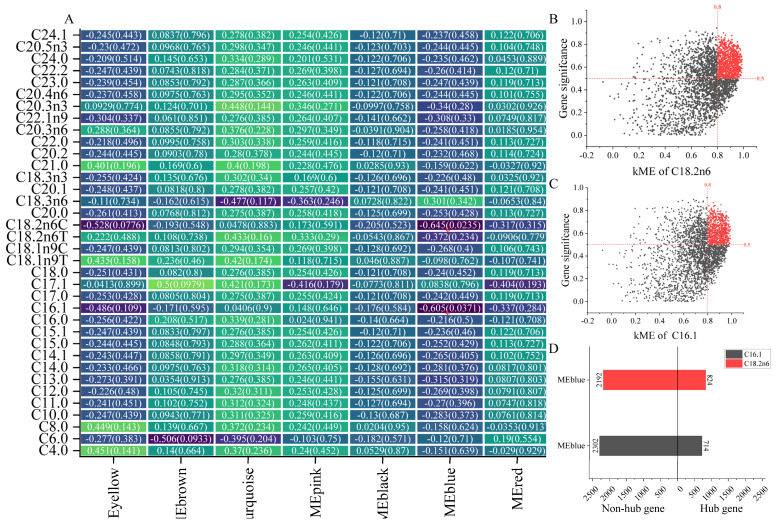
Identification of characteristic modules and hub genes concerning fatty acids. (**A**) Correlation thermogram between 37 fatty acids and gene modules, yellow represents positive correlation, purple represents negative correlation. (**B**,**C**) Scatter plot of hub gene screening of C16:1 and C18:2n6, respectively, in which the X axis was the connectivity of characteristic gene modules and the Y axis was the gene significance; (**D**) summary histogram of genes in the characteristic module of fatty acid. * Indicates that the significance of correlation coefficient is *p* < 0.05.

**Figure 5 ijms-25-09089-f005:**
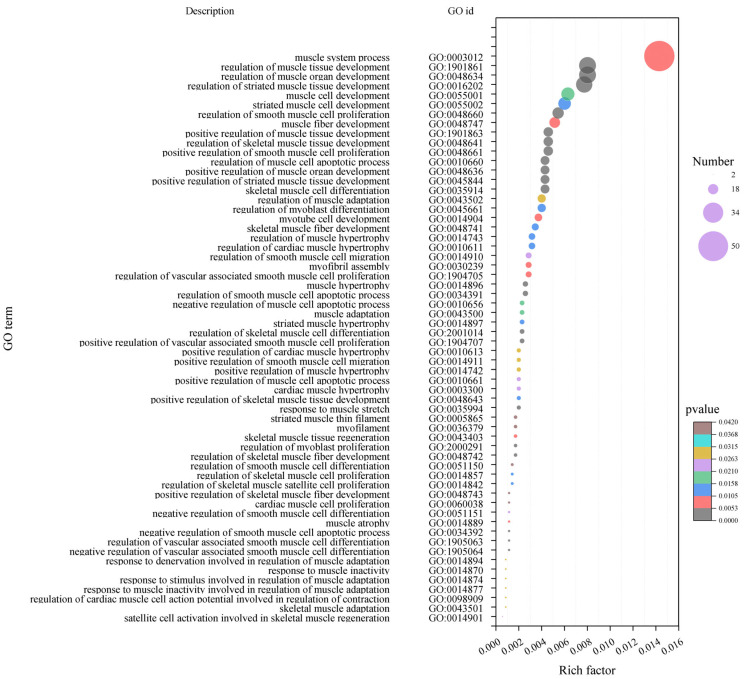
GO enrichment bubble plot of hub gene-related muscle development in nursing period (90 days after birth). The size of the bubble indicates the number of transcripts enriched in the term, and colors indicates the significance of enrichment.

**Figure 6 ijms-25-09089-f006:**
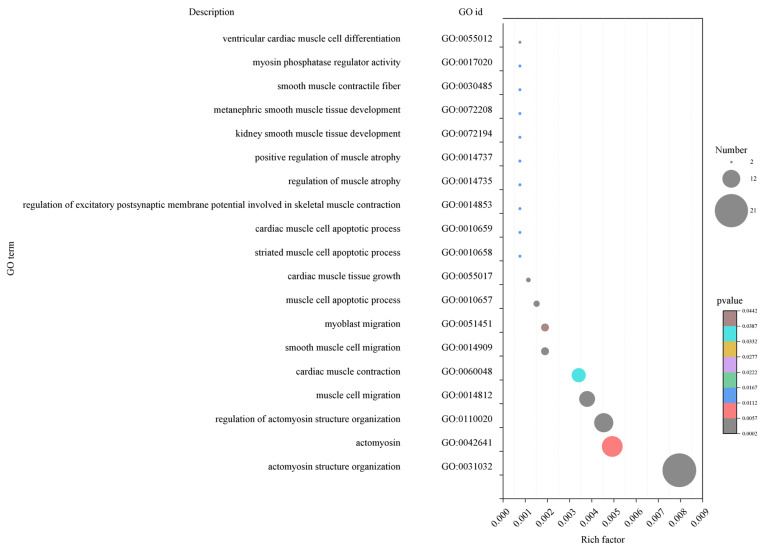
GO enrichment bubble plot of hub genes related to muscle development in early fattening period (150 days after birth). The size of the bubbles indicates the number of transcripts enriched in the term, and colors indicate the significance of enrichment.

**Figure 7 ijms-25-09089-f007:**
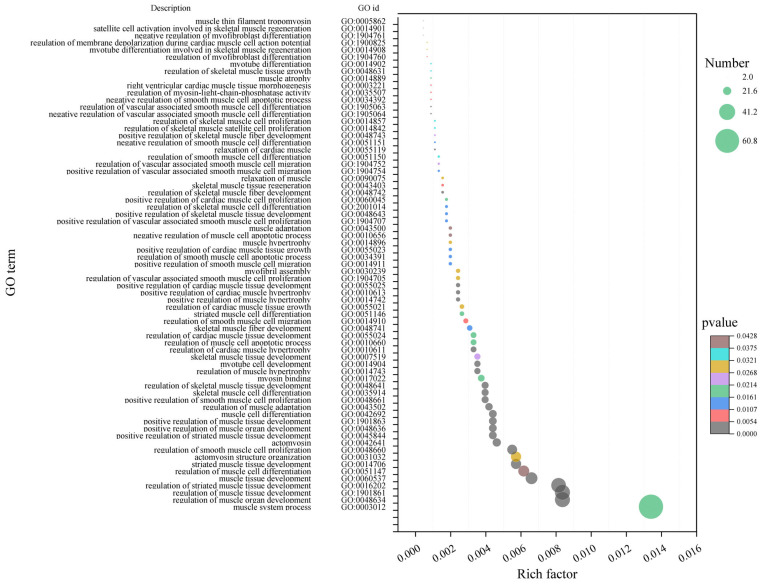
GO enrichment bubble plot of hub genes related to muscle in late fattening period (210 days after birth). The size of bubbles indicates the number of transcripts enriched in the term, and colors indicate the significance of enrichment.

**Figure 8 ijms-25-09089-f008:**
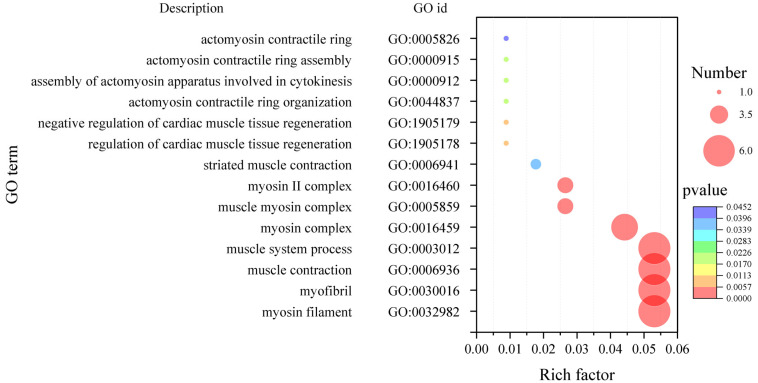
GO enriched bubble plot of hub genes concerning muscle fiber density. The size of the bubbles indicates the number of transcripts enriched in the terms, and colors indicate the significance of enrichment.

**Figure 9 ijms-25-09089-f009:**
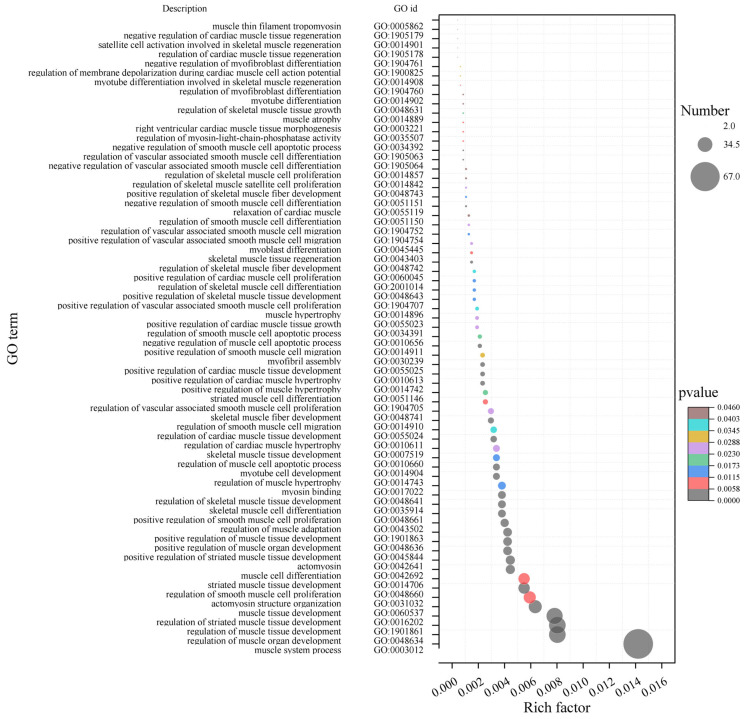
GO enriched bubble plot of hub genes related to muscle fiber diameter. The size of bubbles indicates the number of transcripts enriched in the term, and colors indicate the significance of enrichment.

**Figure 10 ijms-25-09089-f010:**
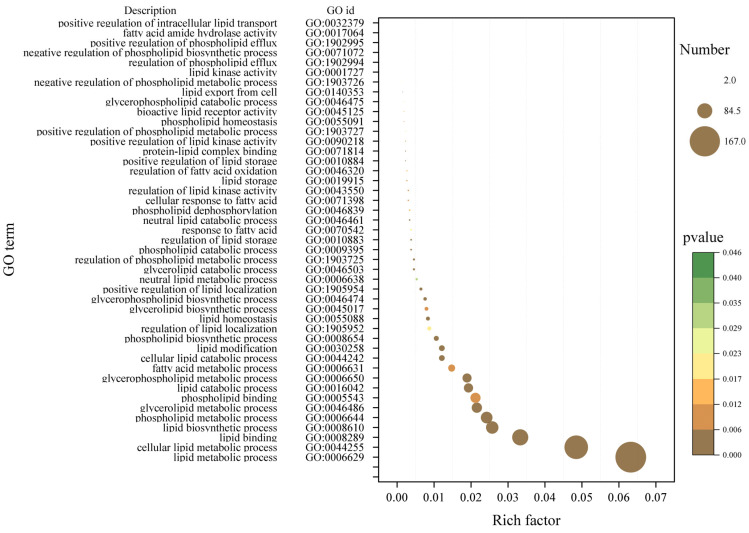
GO enriched bubble plot of hub genes related to lipids. The size of bubbles indicates the number of transcripts enriched in the term, and colors indicate the significance of enrichment.

**Figure 11 ijms-25-09089-f011:**
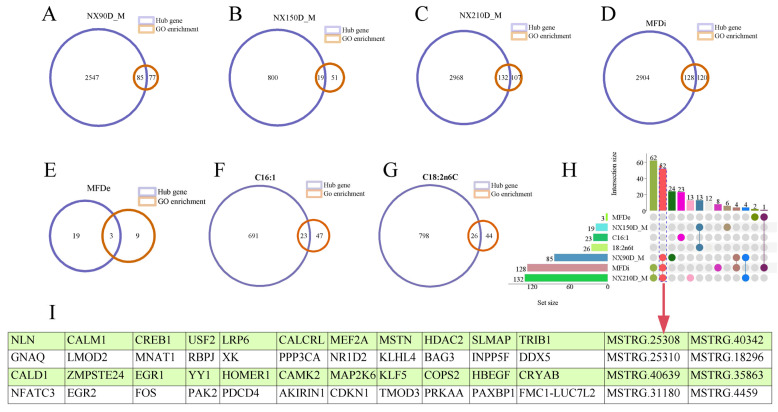
Venn analysis of muscle-related transcripts enriched by WGCNA and GO. (**A**–**G**) Nursery period, early fattening period, late fattening period, muscle fiber diameter, muscle fiber density, C16:1 and C18:2n6C. Upset analysis of transcripts of seven muscle-related phenotypes by WGCNA and muscle-related GO. Upset analysis of muscle-related transcripts in development stage, muscle fatty acid, muscle fiber diameter and density (**H**). * Common muscle-related transcripts representing 90-day-old, 210-day-old and muscle fiber diameter. (**I**) Fifty-two overlapping genes of nursery period, late fattening period and muscle fiber diameter.

**Figure 12 ijms-25-09089-f012:**
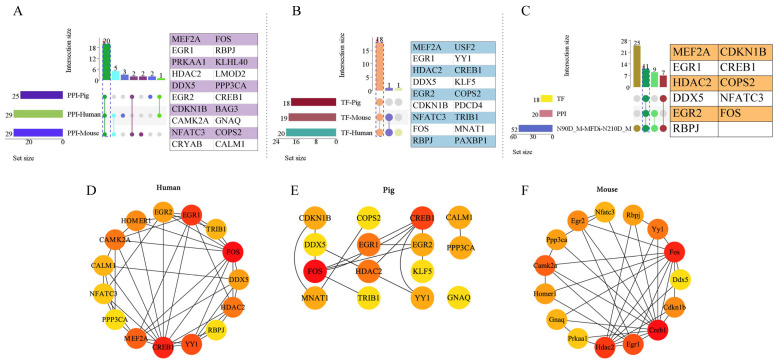
PPI and transcription factor prediction analysis of 44 known genes. Upset analysis of PPI (**A**), transcription factors (**B**) of 44 genes in human, pig and mouse and 52 common hub genes related to muscle development (**C**). Network plot of top15 interaction protein in human, pig and mouse (**D**–**F**), dark red indicated the core gene of interaction. * Represents common transcription factors and interacting proteins in human, pig and mouse.

## Data Availability

Data is contained within the article and Appendix A.

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
