# Peer review of "Exploring the Molecular Mechanism of Skeletal Muscle Development in Ningxiang Pig by Weighted Gene Co-Expression Network Analysis"

_ijms, 2024, doi:10.3390/ijms25169089_

Round 1

Reviewer 1 Report

Comments and Suggestions for Authors

This manuscript is a catalog of the genes expressed in Ningxiang pig, which are identified by WGCNA (probably weighted gene coexpression network analysis) using the samples previously reported but not explained in the manuscript. The description of the Materials and Methods is inadequate, and many abbreviations are not defined, making it difficult to understand what was investigated in this study. The bioinformatics analysis listed some genes, but the image resolution is not good and the font size in the figures is very small to read and understand the results. Supplementary Excel files were broken and could not be opened. The title states “exploring the molecular mechanism of skeletal muscle development”, but the main text does not provide the molecular mechanisms. It is not clear what the novelty, originality, scientific significance, or unique impact of this study is.

Comments

1.          Figures: It’s very hard to read the labels in the figures. The font size should be increased.

2.          Section 2.1: Before describing the results, the type of samples, experimental conditions, methods, and etc. need to be briefly explained, even though it is described in the Materials and Methods. It is hard to understand what the authors analyzed.

3.          Line 351: The experimental samples should be described, not only by citing previous work.

Minor points

4.          Line 16: “WGCNA” should be defined.

5.          Line 40: “LD” should be defined.

6.          And many abbreviations should be defined.

Reviewer 2 Report

Comments and Suggestions for Authors

The manuscript ijms-3117345 is interesting and has the ambitious objective of following the modulation of expression and therefore the entire transcriptome and the consequent morphological and metabolic modifications during development with particular reference to muscle tissue.

Some critical issues do not allow the manuscript to be accepted in this form.

Major revisions:

a) The introduction section needs to be profoundly revised, expanding it, increasing the rationale, the purpose of the research, and the reason to focus on the Ningxiang Pig in particular.

b) Another critical point is the choice (to be explained) of using only one muscle.
The use of a single muscle is very reductive and furthermore does not allow the identification of any specific module/marker common to the muscle tissue.
Considering the heterogeneity of the muscles present, the use of a third muscle would not be excluded in order to have a predominantly oxidative tissue, one with a predominantly glycolytic and a mixed one.

Minor revisions:

a) Completely revise the abstract, also restructuring it with a background.

b) increase the quality of the figures.
For example, in figures 2 and 3 panel A the writings are illegible.
Even in other figures the writing is too small.

c) On line 368 the AA's analysis is cited. Which method? Which tool?

d) Same thing the next line. Even if you cite a method already used, specify the method of fatty acid analysis.

Comments on the Quality of English Language

There are some inaccuracies in the manuscript. I recommend a general linguistic review.

Round 2

Reviewer 1 Report

Comments and Suggestions for Authors

The authors revised the manuscript according to the reviewer's comments.

Author Response

We are deeply grateful for your comments on the manuscript (ijms-3117345) again.

Reviewer 2 Report

Comments and Suggestions for Authors

The authors have sufficiently answered most of the critical issues raised.
However, there are still some unresolved issues that need to be clarified.

Major revision.

The work is well constructed and supported by many qualitatively important experiments.
However, the rationale for the work is weak. Or at least it is poorly explained.

With lines 11-13 and 40-42, what do the authors mean? Discovering molecular determinants so as to genetically modify animals to make them more commercial?
I don't think it is ethical.

Minor revision.

a) The quality of figure 1 needs to be improved. The numbers are not legible.

b) The reference to the instrument and software used for the determination of amino acids is missing.

c) The reference to the instrument and software used for the determination of free fatty acids is missing.

e) Resolve the reference errors on lines 136, 153, 156, 443 etc...

Round 3

Reviewer 2 Report

Comments and Suggestions for Authors

The table in figure 1 is still unreadable.

The numbers entered are unreadable.

Improve the quality of figure 1.

Author Response

comment 1: The table in figure 1 is still unreadable. The numbers entered are unreadable. Improve the quality of figure 1.

Response: Thank you very much for your important comments. We have revised it according to your comments. Please check Figure 1 in the manuscript.